

# A mathematical representation of microalgae distribution in aridisol and water scarcity

Abdolmajid Lababpour[1]

[1]Department of Mechanical Engineering, Shohadaye Hoveizeh University of Technology, Dasht-e Azadeghan, 64418-78986, Iran

*Correspondence to*: Abdolmajid Lababpour (lababpour@shhut.ac.ir)

**Abstract.** The restoration technologies of biological soil crust (BSC) in arid and semi-arid areas can be supported by simulations performed by mathematical models. The present study represents a mathematical model to describe behaviour of the complex microalgae development on the soil surface. A diffusion-reaction system was used in the model formulation which incorporating parameters of photosynthetic organisms, soil water content and physical parameter of soil porosity, extendable for substrates and exchanged gases. For the photosynthetic microalgae, the dynamic system works as a batch mode, while input and output are accounted for soil water-limited substrate. The coupled partial differential equations (PDEs) of model were solved by numerical finite-element method (FEM) after determining model parameters, initial and boundary conditions. The MATLAB features, were used in solving and simulation of model equations. The model outputs reveals that soil water balance shift in microalgae inoculated lands compare to bare lands. Refining and application of the model for the biological soil stabilization and the biocrust restoration process will provide us with an optimized mean for biocrust restoration activities and success in the challenge with land degradation, regenerating a favourable ecosystem state, and reducing dust emission-related problems in the arid and semi-arid areas of the world.

## 1 Introduction

The restoration processes of biological soil microorganisms are significant activities to support soil biological resources and their functions, especially in the arid and semi-arid areas to improve further agricultural activities and reduction of environmental disasters (e.g. (Ci and Yang, 2009; Lababpour, 2016; Rossi et al., 2017)). Biological restoration methods have promising features in environmental sustainability and improving simultaneous functions such as soil fertility improvement, increasing organic and inorganic soil nutrient, increasing soil water capacity, and soil stability against dust storms (e.g. (Bowker, 2007)). However the biological restoration technologies are scarce, as these technologies are complex, difficult to perform, undeveloped or costly. In addition, the success of engineered restoration processes in the natural environments depends on various complex biotic and abiotic interacting factors, which mostly are uncontrollable especially in the landscape practices (O'Donnell et al., 2007; Pointing and Belnap, 2012).



Modelling of biocrust photosynthetic microorganisms and their interaction with soil physical characteristics and climate factors
seems particularly interesting (e.g. (Albano et al., 2017; Kinast et al., 2016; Vasilyeva et al., 2016)), which provide information about involved processes and mechanisms, experimental data are not available for now. However, modelling of the system is difficult as it depends to various type and scale processes and variables such as microorganism's type and population, water and nutrient availability and weather conditions. For soil water modelling purposes, effective factors could be considered are rainfall and irrigation, evapotranspiration, percolation and water runoff, as well as water consumption by microorganisms for
their physiological activities. For the microorganisms modelling purposes, temporal and spatial distribution on the soil surface, their growth and death rate, light and nutrient availability are among effective factors.

As the biocrust structure have similarities to the solid state bioreactor systems (Larios-Cruz et al., 2017) especially those developed for biofilm (see Table 1), therefore, insights into possible mathematical strategies can be gained from studies
undertaken in the context of biofilm systems in development of biocrust models (Alpkvist et al., 2006; Boltz et al., 2010). However, most biofilm models developed so far include a highly simplified description of bed (soil) particle characteristics and moisture dynamics. This approach has the disadvantage that could not predict soil physical properties and soil water balance required for soil restoration. Hereinafter, to improve the description of soil physical properties and soil water balance, we included the soil porosity in the model which among the soil physical characteristics, is fairly well standardized in definition
and measurement techniques. In addition, porosity can be related to other soil physical parameters such as hydraulic conductivity, pore size and particle size (Nimmo, 2004). In addition, we considered horizontal distribution of biomass across the soil surface, which rarely considered in biofilm modelling approaches.

Table 1 A comparisons of soil biocrust and phototrophic biofilm (Rossi et al., 2017).
Taken together, the reports in literature reveals that biotic and abiotic parameters have important role in model predictions used for restoration of soil in arid lands. In this framework, here we concentrate on the soil water and biomass of photosynthetic microorganisms which are critical in soil restoration in the arid areas with water scarcity. The emphasis of this paper lies on describing the temporal, spatial and kinetic parameters in the model, that is, model components, processes, parameters, and underlying assumptions. This is followed by a comparison of experimental and model estimation data of cyanobacteria
*Microcoleus* cultures inoculated on the soil surface. The model was considered for further development of biological soil crust restoration, especially in the arid and semi-arid areas by considering soil particle size, porosity (especially structural porosity (Nimmo, 2004)) and density, which affect water and nutrient availability. As there are many challenges before model can be used in practice, this study is open for several further work are required to investigate the simulation of real situation of BSC systems to combat desertification.



## 2 Model formulation

### 2.1 Domain and characteristics of the model

As cyanobacteria cells grow and mature, they take on differentiate forms, properties and functions, mainly in a thin layer of few millimetre very close to the soil surface. They absorb sunlight and use it in photosynthetic mechanisms for cell physiological activities. The nutrient will be up taken in water soluble form from soil porespece filled with water and gases. Therefore, available water has a critical role for cell physiological activities, in addition to the gas exchange between the cells and environment. These essential features are to be captured in the mathematical model.

For simplicity, BSC volume between two finite imaginary flat parallel plates oriented horizontally were assumed as model domain (Fig.1). We ignore for the present the geometric details of the domain surfaces and any thickening occurring along the length and width of the surfaces through aging. The BSC constituents are changed between stationary plates based on their gradients. The movement of system components within the plate happen by diffusion rather than convection (low convection per pass, ≪ 5%). Therefore, it creates a plug flow behaviour in reaction zone. Then, the distribution pattern of biomass, water and nutrients have been investigated by diffusion reaction equation.

Functional features and variable properties are characterized as functions of the plate length and width coordinate pair $(x, y)$ in Cartesian coordination system. In the other word, we considered $u (x, y, t)$ be the variables such as biomass of a point $(x, y)$ ∈ $\Omega$ at time $t$, where $\Omega \subset R^3$ is a domain. The $u (x, y, t)$ satisfies in $\Omega \times [0, Lx) \times [0, Ly) \times [0, Lz)$ and then, domain may be represented as

$$\Omega := (0, L_x) \times (0, L_y) \times (0, L_z) \subset R^3 \tag{1}$$

In our numerical implementation, the squared domain geometry and mesh were specified by a matrix of points, where the number of points are discretized into the 679 nodes and 1272 triangles.

We have to choose variables that reflect the specificity of biocrust in the selected domain. From the literature we found that commonly, biomass concentration is considered to be included into the model due to its important effects on soil characteristics. In describing the biocrust development on the soil surface, it will also be logical to take into account the water role carrying nutrient for microorganisms, which is very important in further soil restoration. In addition, the water concentration on biomass growth is quite an important, and sometimes is determining factor of the restoration processes in the desert areas with water scarcity. Hence, we model the cell growth ($B$), and soil water uptake ($w$) as an interdependent mass-balance system. The temporal and spatial development of biomass and water movement variables are described by mass-balance partial differential equations. Later, new variables can be added into the model or excluded.

Any of the variables were shown with a mass balanced PDE equation in the form of diffusion-reaction equation. The variables and their interactions in each equation terms were then explained with auxiliary equations to include other factors such as light illumination. Afterward, the unit consistency in the equations were evaluated. The biocrust formation is modelled for ten days based on the experimental data.





**Fig. 1 A schematic assumed domain geometry.**

For simplicity, a single filamentous cyanobacteria species assumed to occupy the domain in xy-plane with the *x (t)* and *y (t)* functions in the bounded cases. A limiting water substrate with concentration *w(x, y, t)* is present throughout xy > 0. The water

containing nutrients diffuses in the horizontal plane, in a thin layer close to the surface while are consumed by cells for growing. In the other word, when water is available, the biomass growing. Conversely, water is consumed when biomass growing. It means we have a coupled system. Therefore, the problem requires the solution of a set of two coupled PDEs with biomass and water availability as dependent variables. In addition, we considered the half space x,y > 0 in R$^3$ with a wall placed at x,y = 0.

The model proposed here describes the development of biocrust with various simplification assumptions. Table 2 show the assumptions used in the development of present biocrust model.

Table 2 Assumptions used in the BSC modelling of filamentous cyanobacteria.

For a given function *F* the general equation of second order for the biomass state variable *B(x, y, t)* is given by

$$F\left(x, y, t, B, B_{xx}, B_{yy}, B_t, \right) = 0,  \tag{2}$$

in $\Omega \in \mathrm{R}^3$. $B_{xx}$ is biomass derivative in the x, and $B_{yy}$ is the biomass derivative in the y directions. Assume that the function *F* is sufficiently regular in its arguments. The mass-balance second order Eq. (2) for the biomass *B* is rewritten as

$$\frac{\partial B}{\partial t} = D_B \left( \frac{\partial^2 B}{\partial x^2} + \frac{\partial^2 B}{\partial y^2} \right) + \mu B - dB,  \tag{3}$$

where $D_B$ is distribution coefficient of biomass in the x and y direction, $\mu$ is specific growth rate, and *d* is cell death coefficient. To include soil porosity, Eq. (3) becomes as

$$nV \frac{\partial B}{\partial t} = nVD_B \left( \frac{\partial^2 B}{\partial x^2} + \frac{\partial^2 B}{\partial y^2} \right) + nV\mu B - nVdB,  \tag{4}$$

The effect of light illumination on specific growth rate, $\mu$ is defined by empirical relationship has been obtained from a study on the photosynthesis of phytoplankton in the sea (DiToro et al., 1971).

$$f(I) = \frac{I}{I_s} e^{1 - \frac{I}{I_s}},  \tag{5}$$

where *I* the instantaneous illumination rate, and $I_s$ are the optimal illumination rate (µmol photon m$^{-2}$ d$^{-1}$). The relationship between growth rate and the concentration of a single growth-controlling substrate (water) is shown by logistic model (Chambon et al., 2013). The relationship of water consumption and growth rates were shown via two parameters of maximum

growth rate $\mu_{max}$ and the substrate affinity constant $K_w$. The link between light illumination and water utilization is represented by logistic model which linearly relate the yield coefficient to the specific growth rate of biomass (Kovárová-Kovar and Egli, 1998).



The inclusion of illumination term in the Eq. (4) implies that the model is applicable for the microorganisms with photosynthetic activities. Biomass production increases as a function of light intensity until an optimal intensity is reached, and beyond that optimal value, production varies in accordance with the type of light source. That is, algal growth curves under condition of continuous light and intermittent light (typically 14 hours of light, 10 hours of dark) are unique and species dependent.

The mass-balance equation for soil water in a horizontal top soil layer (biocrust) can be described by the following equation

$$nV\frac{\partial w}{\partial t} = nVD_w\left(\frac{\partial^2 w}{\partial x^2}+\frac{\partial^2 w}{\partial y^2}\right)+R(t)+I(w(t))-ET(w(t))-LQ(w(t))-nV\frac{1}{Y_{\frac{B}{w}}}(\mu-d)B \qquad (6)$$

The terms in this equation from the left are for temporal $nV\,\partial w/\partial t$ and spatial $nVD_w\,(\partial^2 w/\partial x^2+\partial^2 w/\partial y^2)$ soil water profile, time $t$, soil water volume $nV$ ($A$ surface area, $L$ length and $n$ porosity), water concentration $w$, rainfall $R(t)$(kg/d), irrigation $I(w(t))$ (kg/d), evapotranspiration $ET(w(t))$(kg/d), the combination of percolation and runoff water losses $LQ(w(t))$(kg/d), and water uptake by photosynthetic microorganisms $nV\mu B/Y$ (kg/d), in which $\mu$ and $Y$ are for specific growth rate and conversion yield (-), respectively (Albano et al., 2017). The term rainfall and irrigation are modelled according to (Bartlett et al., 2015), the actual evapotranspiration is estimated for arid and semiarid land areas by empirical data and the relationship described as $Et_a + ET_p = 2ET_w$, in which $Et_a$ is actual evapotranspiration, $ET_p$ is climatic parameter, and $ET_w$ is wet environmental evapotranspiration, soil water losses by percolation and runoff are modelled according to (Bartlett et al., 2015), and soil water consumption by photosynthetic microorganisms are represented by logistic equation multiplied by yield conversion coefficient, $Y_{w/B}$ (Albano et al., 2017; Baudena et al., 2013). A square 16 m$^2$ was selected in solving the equations.

With no water input and output assumption from rain, irrigation, percolation and evapotranspiration, Eq. (6) reduce to Eq. (7) as

$$nV\frac{\partial w}{\partial t} = nVD_w\left(\frac{\partial^2 w}{\partial x^2}+\frac{\partial^2 w}{\partial y^2}\right)-nV\frac{1}{Y_{\frac{B}{w}}}(\mu-d)B \qquad (7)$$

For a constant soil porosity, we can assuming homogeneous bed, and then Eq. (7) reduces to

$$\frac{\partial w}{\partial t} = D_w\left(\frac{\partial^2 w}{\partial x^2}+\frac{\partial^2 w}{\partial y^2}\right)-\frac{1}{Y_{\frac{B}{w}}}(\mu-d)B \qquad (8)$$

This equation shows temporal and spatial soil water in the x and y directions because of diffusion, and its consumption by microorganisms. Various soil types porosity of arid areas can be included in the model through parameter $n$, which is commonly in the range of 0.3 – 075 (Ci and Yang, 2009). It is also empirical relations are available may be used for converting soil porosity to other soil physical characteristics such as particle size and density (Foster and Miklavcic, 2013).

The initial conditions associated with the biomass equation and water solution is defined as

$$B(x,y,0)=B_0(x,y)=0.02, \qquad x\in\Omega, \qquad (9)$$
$$w(x,y,0)=w_0(x,y)=p, \qquad x\in\Omega, \qquad (10)$$





where $B_0$ and $w_0$ are defined in all the plate at t = 0 and $p$ is average fraction of total available soil water, which is in the range of $0.1 < p < 0.8$. Based on the experimental data, the initial biomass concentration of 0.02 kg/m³ and soil water content of $p =$ 0.2 were considered as initial conditions. This show that all of domain points have similar conditions of biomass and water content in the beginning.

The boundary conditions for the system shown by Eqs. (4), and (7) are given by

$$B(x, y, t) = w(x, y, t), \qquad x \in \Omega, \qquad t \geq 0 \tag{11}$$

where $B(x, y, t)$ and $w(x, y, t)$ are given as Dirichlet bounded boundary conditions is imposed between horizontal coordinates $0 < x < 4$ and $0 < y < 4$ m. The boundary conditions are equal to initial conditions at boundaries.

Gathering the above PDE equations, we have the model incorporated two 2-D PDE equations, 2 initial conditions, 8 boundary conditions, and 8 parameters divided to biomass and soil water solution related variables in a same domain. The involved model parameters estimated from experimental data or literature are summarized in Table 3.

Table 3 Summary of parameters used for the computation of model and their characteristics.

## 2.2 Numerical model solution algorithm, codes and software

The numerical solution of nonlinear system of coupled partial differential equations represented by Eqs. (4), and (7) were provided using the Matlab software version 8.3.0.532 (Matlab 2014a, USA). We used the parabolic and app toolbox packages for PDEs solution. In the numerical scheme, the time derivative with respect to $t$ and space partial derivative with respect to $x$ and $y$ in Eqs. (4), and (7) are discretized to a consistent order of approximation using central difference. Equations (4) and (7) were written in matrix coefficient form and solved to find the biomass and water solution horizontal profile in each time and space point on the soil surface. For each time step, the specific growth rate, $\mu$, was calculated from light function $f(I)$, and soil water function $f(w)$, in each element from the previous time space. These specific growth rate values then used to calculate biomass and soil water concentration in the current time step. The solver uses finite element method (FEM) for solving partial differential equations (Boltz et al., 2010). The program implication was followed in 6 steps of geometry definition, set the initial and boundary conditions and setting time's steps, create the PDE coefficients, mesh creation, solving by solver and plot the solution results.

## 3 Results

### 3.1 Total biomass

The developed mathematical model is able to predict concentration profiles for biomass, soil water as substrate, and light profiles in the biocrust as well as carbon dioxide concentrations and solute nutrient. Figure 2 shows a typical time course of a single simulation biomass profile for the biocrust. According to Fig. 2, the cyanobacteria biomass concentration increase as





time pass during cultivation period close to the soil surface, which is in accordance with experimental observations in the petri dish cultures (Lababpour and Kaviani, 2016). Although the spatial distribution of biomass is not homogeneous in real soil conditions, as it is function of variables such as soil non-homogeneous structure and behaviour of microorganisms, the model

show uniform biomass distribution in constructed biocrust as expected. A relatively homogeneous distribution on the surface can be observed in experimental biocrust set up. This difference is shown to be practically are not significant. However, the approximately close biomass production appears by the model is an expectable observation under the conditions of simulation. Besides the uncertainty caused by additional model parameters used in the biocrust model, there also be an added uncertainty caused by the model structure and simplification assumptions is not considered in this study.

The increasing biomass within the biocrust stems from a reaction-diffusion interaction. The model predicts the biomass concentration gradient as well as the gradient of soil water concentration. Biomass concentration is increasing near the biocrust interface and becomes reduced in the biocrust interior. However its distribution is homogeneous in the domain x and y directions surface. The profile of viable cell fraction is reversed in which cell numbers is in higher numbers of the biocrust surface and are decrease effectively near the domain substratum. Viable cell numbers and biocrust thickness reduction were

dissimilar measures of light efficacy. The general behaviour predicted by the model is in good qualitative agreement with an experimental observations. As the initial seeding was considered similar in domain, therefore, biomass distribution would be homogeneous in all of the surface.

Fig. 2 Results of the temporal and spatial biomass concentration distribution profile in the x and y surface. (a) Temporal distribution, and
(b) spatial distribution with surface plot.

The modelling of soil water is more complex as it is not only dependent to up taken by microorganisms, but also dependent to forces acting on the soil water mass balance such as soil porespaces, evapotranspiration, and input-output water flow to soil. If we assume no input-output water flow, and water is free for movement in the soil, the model prediction reveals of a

logarithmic water reduction curve during cultivation period. The decreasing speed in this case is most inversely corresponds to cell growth rate. Figure 3 reveals time profile of soil water in the domain during cultivation period in the horizontal surface.

Fig. 3 Results of the soil water concentration distribution profile in the x and y surface. (a) Temporal distribution, and (b) spatial distribution with surface plot.


## 3.2 Potential of model for water estimation in biocrust restoration

In order to assess the potential of using model for calculating of water required for biocrust restoration, the involved irrigation and rainfall, evapotranspiration, percolation and up taken by microorganisms terms are considered in Eq. (7). Mass balance Eq. (7) could predict the water forms involving in soil restoration system. By measurement or approximate estimation of



equation terms, the water need, lost water and the net water used for soil restoring microorganisms are estimated. With such data, we can calculate the required water for used species per hectare covered per month to growth cells in optimal condition, which still remain unclear in arid areas. The prediction of soil water initial condition is also critical for subsequent restoration progress economically.

## 4 Discussion

The main objective of this study was providing a model to be used for the prediction of growth of photosynthetic microorganisms on the arid and semiarid soil surface to make insights for field restoration studies in the aridisol lands. In addition, we focused to find interrelationship of biomass and soil water dependent variables through solving a set of two coupled PDEs. To formulate model, two factors of simplicity and accuracy was considered. Therefore, a PDE equation

representing the growth of cells as biocrust and another for soil water consumption by cells were developed.

Our 2-D horizontal model complements the models who considered the 1-D vertical biomass growth on the soil surface. As biocrust horizontal patterns are more important than vertical features to soil surface functions, we have focused on what happen on the aridisol biocrust. Based on the vastly reported results of the effects of light illumination on the growth of photosynthetic microorganisms, in this study, an attention has been done on light intensity in the biocrust model of photosynthetic

cyanobacteria. To this purpose, the empirical relation represented by DiToro et al was used which account for photoinhibitory effect on the cell growth. The DiToro model is also applicable in study the effect of light illumination has inhibitory effects on the growth of various photosynthetic microorganisms (Picioreanu et al., 2004; Samsó and Garcia, 2013).

Apart from the light intensity function, mentioned in the model, the proposed biocrust model included major processes of substrate consumption, cell growth and biomass production, initial biomass inoculum and atmosphere parameters, which are

main practical parameters in biocrust restoration on the soil surface. However, it does not included several challenges in engineering applications of biocrust restoration. Some of the current challenges are faced in proposed model are the lack of fractal biocrust growth rate on the soil surface coverage, exopolysaccharide (EPS) release by cyanobacteria cells, the attachment and detachment of biocrust to the soil surface and capture of soil particles inside of the biocrust matrix, morphology of biocrust growth and filaments, model coefficients, 3-D modelling, regrowth after drying, production of organic nutrients by

cells in the soil applicable by other microorganisms, estimation of long-time behaviour of cyanobacteria biocrust, and ecological influence of cyanobacteria in the soil community. These limitation indicate the requirements to further development in the biocrust modelling field (Samsó and Garcia, 2013). We plan to include EPS production sub-model and functions in the future study.

The model described in this paper is intended to be a first step in modelling of arid land biocrust, in particular to investigate

the conditions necessary for water irrigation and inoculation to succeed soil restoration. The generic modelling approach we





have adopted aims to describe early biocrust development on the soil surface in arid and semiarid areas. Extensions to the model can be support the inoculation of soil surface in largescale to combat desertification.

Although various simplification assumptions were applied in model development, but the use of model sheds light on the biological process of biocrust formation since it simulates some central issues of biocrust including interrelations of soil water profile and horizontal biomass distribution. The 2-D biomass, soil water solution and light intensity modelling make a more realistic feature compare to other modelled without light factor.

Validation of the model was performed using experimental *Microcoleus* biocrust previously developed on the soil surface (Lababpour and Kaviani, 2016). However, only biomass growth and spread of surface coverage were controlled in laboratory experiments. The model prediction was in agreement with the experimental observations (Lababpour and Kaviani, 2016). In this research we used Entisol order soil samples of Khuzestan for model calibration, which are sub-orders of Aridisol in the soil taxonomy studies. However, as there are thousands soil series with different features, it is required to calibrate the model with various soil series and taxonomies especially Aridisol types, for closing model applicability to practical soil restoration performance.

The simple proposed model consider only two dependent variables in the biocrust to prevent model complexity. However, generally, biocrust consist of interwoven community of various microorganisms in a variable climate. To further application of biological soil crust models in the field studies, they may include various biocrust constituents has been reported partly in literature such as for lichen and mosses (Samsó and Garcia, 2013). Different solutions have been proposed so far, all of them still needing much improvement in order to generate a realistic picture of biocrust models. It is hoped that the results of this paper will provide a basis for the development of more sophisticated models and will provoke a variety of experiments, directed at the quantitative understanding of biocrust growth and soil restoration in arid and semiarid areas (*e.g.*, (Vidriales-Escobar et al., 2017)).

# 5 Conclusions

This study present a cyanobacteria-soil water interaction model, based on the biofilm models, assuming the cyanobacteria biomass and available water in soil are state variables. The objective was to develop a simple model to represent the cyanobacteria - soil - climate dynamics in arid and semiarid zones such as southwest area of Iran. However, modelling is difficult as it depends to multi-direction interaction between effective parameters. It was estimate some of the general biocrust specifications developed by model which have been evaluated experimentally. These included the biomass and soil water profile inside of the biocrust as a function of space and cultivation time. The model estimations were support the empirical biocrust growth observations. The model would be applicable for simulation of other factors influencing biocrust restoration such as temperature and the fate of the biocrust in aridisol, by including the related sub-model equations. The sensitivity measurements did not performed which is in progress for further model quality improvements. Within the studied simplified domain, the model suggest higher biomass productivity if the water and light are adequate for the cells. Several improvements



are required to be included such as seasonality before the model can be used for practical applications in the conditions usually demonstrated in the arid and semi-arid areas.

**Acknowledgments**

The author would like to thanks Shohadaye Hoveizeh University of Technology for their support. I would also like to thank anonymous reviewers for their constructive suggestions that helped improving the manuscript.

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



**Figure captions**

Fig. 1 A schematic assumed domain geometry.

Fig. 2 Results of the temporal and spatial biomass concentration distribution profile in the x and y surface. (a) Temporal distribution, and

(b) spatial distribution with surface plot.

Fig. 3 Results of the soil water concentration distribution profile in the x and y surface. (a) Temporal distribution, and (b) spatial distribution with surface plot.

**Table captions**

Table 1 A comparisons of soil biocrust and phototrophic biofilm (Rossi et al., 2017).

Table 2 Assumptions used in the BSC modelling of filamentous cyanobacteria.

Table 3 Summary of parameters used for the computation of model and their characteristics.



Table 1

| BSC | Biofilm |
|---|---|
| Has higher species diversity | Has single or few species diversity |
| The associated communities are bacteria, fungi, lichen and mosses | The associated communities are similar microorganisms e.g. bacteria |
| In addition to cyanobacteria, other microorganisms such as fungi participate in EPS production. | Cyanobacteria are main EPS producers |
| Coral and fine particles are main components | Only biofilm components are constituent species |
| They are more porous | They are less porous |
| *Microcoleus* lack UV protecting pigments, then stratifies in the sub-surface, while some other such as *Nostoc* having UV screening pigments stratify at the soil surface. | Relatively homogeneous |
| The substratum cannot be defined | The substratum is fixed flat plate |
| There is no detachments, and the decayed cell remain in the soil. | The cell detached from the biofilm surface. |






Table 2

| Assumed condition | Real condition |
| --- | --- |
| Two-dimensional (2D) - BSC structure parallel to soil surface | Three-dimensional (3D) biocrust structure |
| Homogenous biofilm structure, (uniformly thick planar aggregate) | Heterogeneous biocrust structure (physical, chemical, biological and geometrical) |
| Biocrust is solid matrix and rigid structure | Biocrust is multiphase system |
| Pure species biocrust | Diverse multispecies biocrust |
| Fixed biocrust boundary conditions | Moving biocrust boundary conditions |
| Biocrust is flat | Biocrust is fractal shape |
| Biofilm thickness is constant. | Biocrust volume and thickness is variable |
| Diffusion rate of substrate is constant | Diffusion rate of substrate is variable |
| Temperature is constant and in optimum conditions | There is temperature fluctuation in the biocrust |
| All variables over areas parallel to the substratum are constant. | Variables are not uniform over surface area parallel to substratum |
| Light kinetics describe by DiToro et al kinetics | Kinetic of light variation is complex inside of the biocrust |
| Substrate concentration and environmental parameters are not limiting except light illumination and water | Many substrate and environmental parameters are limiting and complex inside of the biocrust |
| Light enters the biocrust domain perpendicular to surface | Light angle and intensity change with time and locations |
| The domain boundaries are no diffusible and fix | The biocrust boundaries are not fix and growing fractal. |



Table 3

| Name | Symbol | Unit | Value | Ref. |
| --- | --- | --- | --- | --- |
| Biomass distribution coefficient | $D_B$ | g/L | 6.25E-4 | a |
| Maximum specific growth rate | $\mu_{max}$ | $d^{-1}$ | 0.4 | b |
| Water distribution coefficient | $D_w$ | g/L | 6.25E-2 | a |
| Optimal light illumination coefficient | $I_s$ | $\mu mol\ photone.m^{-2}.d^{-1}$ | 0.007 | a |
| Michelis – Menton coefficient | $K_w$ | Kg C/$m^2$ | 0.01 | b |
| Soil porosity | n | % | 0.3-0.7 | c |
| Soil water volume | V | $m^3$ water/ $m^3$soil | 0.2 | This study |
| Time | t | d | | This study |
| Soil water concentration | w | kg/$m^3$ | | This study |

a, (DiToro et al., 1971); b, (Systems, 2009); c,(Schiavone, 2016).





Fig. 1

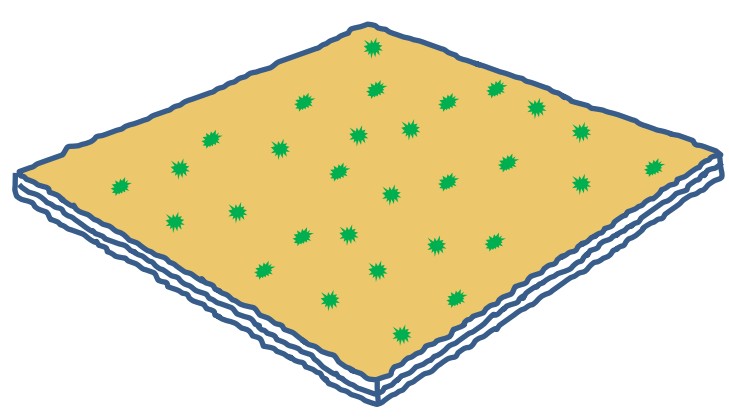





Fig. 2

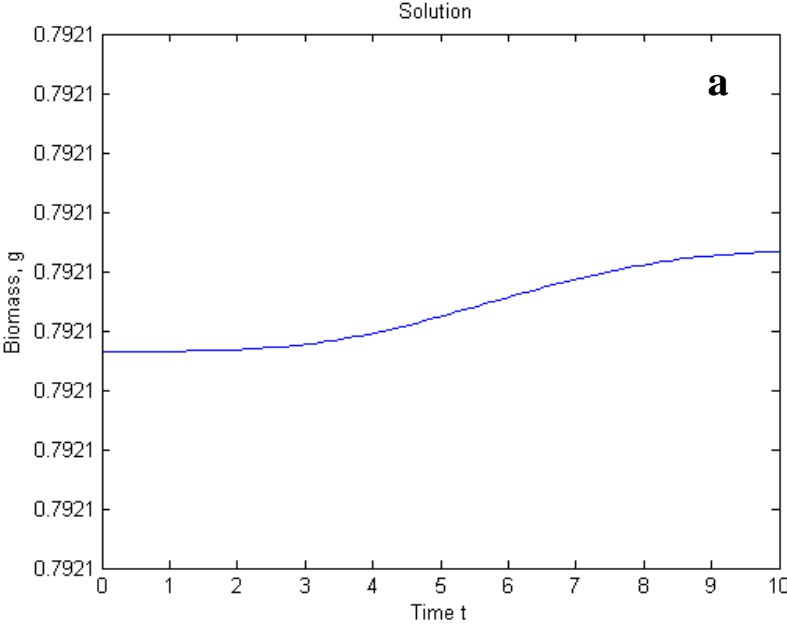

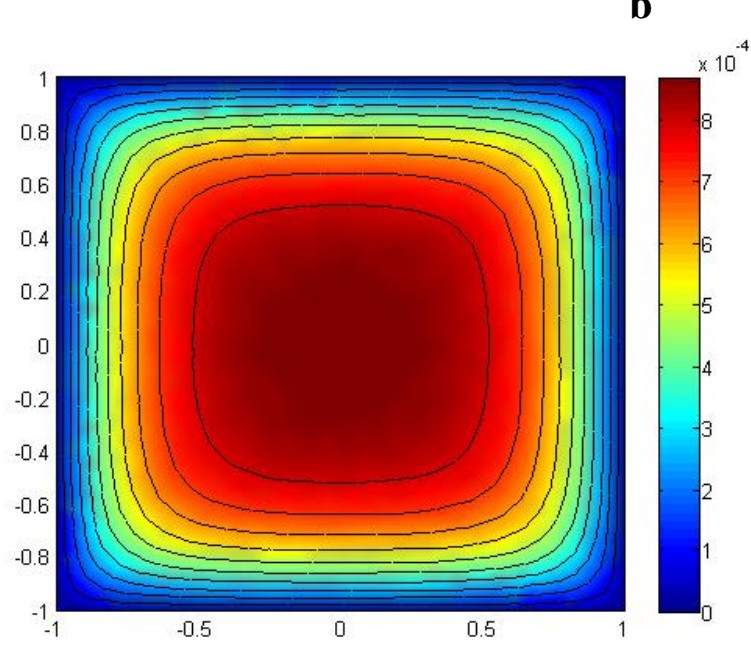





Fig. 3

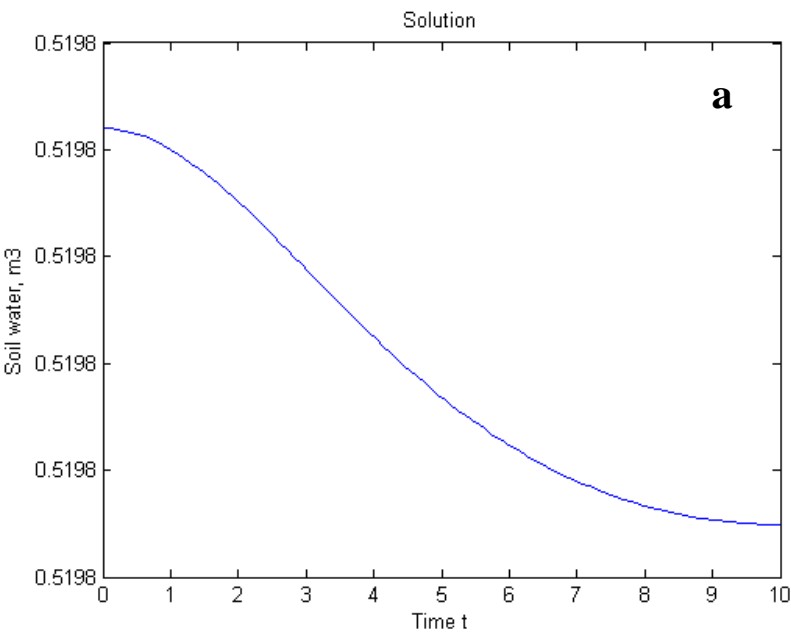

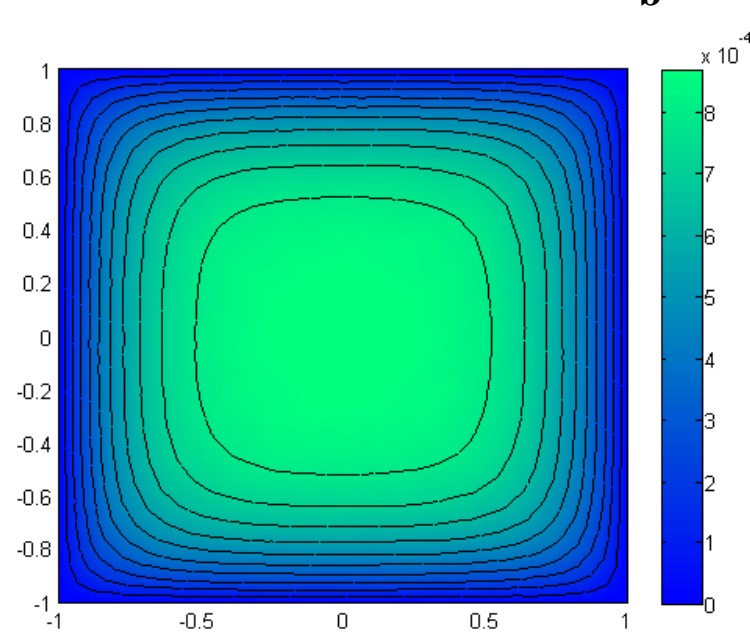