# Peer review of "A mathematical representation of microalgae distribution in aridisol and water scarcity"

_Biogeosciences, 2017_

## Referee Comment (RC1) · Anonymous Referee #1 · 24 Oct 2017

In the manuscript entitled "A mathematical representation of microalgae distribution in aridisol and water scarcity", Lababpour presents a mathematical model of microalgae development on surfaces. The presented model attempts to describe the interaction between cyanobacteria and soil water with the possible inclusion of climatic variables. Although the approach of using a reaction diffusion equation of biomass and water is interesting, the manuscript does not provide in-depth tests to grasp the applicability and predictability of the model. Furthermore, the presented equations raise some questions regarding the model capability that author discusses. I enjoyed reading about the work, but have several major issues with the model and its presentation for the publication.

Firstly, the author describes the growth of cyanobacteria to be a function of light intensity (which should be time dependent, day and night) and water. However, the growth term is not well described in the manuscript and the solution does not reflect such behaviour.

Secondly, unlike the title highlights that the model is for aridisol, the mathematical description does not include any of the properties of such soils. Using water as one of main variables and introducing the multiplication of the porosity $n$ might imply such application to dry soils. However, the coupled PDE only replaces the limiting nutrient to water in the conventional reaction diffusion equation for microbial growth. Especially, without including the input and output of water in the equation, the role of soil physical property (porosity in this work) cancels out, thus no soils in the equation. To be constructive, using water content instead of water concentration (it is quite difficult to understand what this term means) can be considerable to include properties of soils. The Richards equation with the saturation based soil water diffusivity can provide the proper water dynamics in dry environments.

Thirdly, the example solution of the model in this manuscript is too simple, even trivial. Simulating without any heterogeneity in the domain and uniform distributions of both variables as initial conditions should result in no differences during the dynamics (as the figures show and the author has mentioned in the result section). Furthermore, I cannot find any physical reasoning for the used boundary condition, equation (11).

Finally, the units provided in Table 3, do not seem to be correct. All units and dimensions need to be checked again in the model.

Unfortunately, the model has major issues and needs further development. The attempt, however, to seek for a simplified representation of such a complex system is greatfully acknowledged.

---

## Referee Comment (RC2) · Anonymous Referee #2 · 7 Nov 2017

**Report on the paper "A mathematical representation of microalgae distribution in aridisol and water scarcity" by A. Lababpour**

**November 6, 2017**

The author presents a model consisting of two diffusion-reaction equations for biomass and water development. The resulting model equations are solved using a Matlab toolbox. The author thoroughly discusses extensions of the modeling approach and outline directions of further research. He particularly discusses influencing factors for the three dimensional development of the soil biocrust. Although this part is quite well explained and the topic itself is of interest, I do not recommend the manuscript in its current stage for publishing in Biogeosciences. For details see the following comments below.

1. The model is very simplified although possible reasonable extensions are discussed. The predictive power is therefore questionable. A quantitative discussion e.g. in comparison with with data should be added.

2. The model under consideration and its precise mathematical formulation remains unclear. The author should clearly differentiate what could be done and what is done within the current research. The model that is implemented it not clearly stated in its whole at any point. Exactly this model should be summarized once in the paper including the domain of definition, the used variables and parameters.

    (a) The geometric setting is unclear. First the three dimensional domain $[0, L_x] \times [0, L_y] \times [0, L_z]$ is discussed. This is thereafter changed to the half space (which actually is not the half space but the first quadrant). Thereafter a square $[0, 4]^2$ is considered.

    (b) It is unclear how the porosity enters the investigations. The equation (4) is reasonable only in case that the porosity is independent of space. Otherwise I would expect that the diffusion terms reads

$\nabla \cdot (nVD_B\nabla B)$. However, it seems that there is no modulation of the porosity throughout the simulation scenario, e.g. in the initial conditions.

(c) The inclusion of the illumination rate into the growth term of equation (4) is unclear. How is $I$ related to $\mu$ and how is it determined/defined in the simulation scenario? Along the same lines $d$ is undefined in (8). Does the logistic model enter the proposed model in terms the growth rate $\mu$ in (4) and (8) and if so how does it?

(d) The statement of boundary condition in (11) makes no sense. First the boundary conditions must be prescribed on the boundary of the domain rather that in the domain itself. Second the variables are independent quantities. I assume the author means $B = 0.02$ and $w = 0.2$ are prescribed on the boundary.

3. The outcome is of the simulations are not convincing:

(a) The scaling in Figure 2 and 3 is unclear. Variations are seen in order $10^{-4}$ or even below. This seems not to be relevant compared to the initial/boundary values or even the reference value of $0.7921$ and $0.5198$ in Figure 2 and 3. The smallness of the variations could even be due to numerical or rounding errors. It is remarkable that the values for $B$ and $w$ are within the same order of magnitude directly although starting with one order of magnitude difference initially.

(b) The initial conditions are not matched for $t = 0$ in Figure 2 and 3. This does not make the results reliable.

(c) It is unclear for which time the spatial distribution is plotted. The values are very small (order of $10^{-4}$) compared to the chosen initial values. Is zero maybe a steady state solution?

(d) Explicit comparison to data is missing. This could maybe shed some light into the results and make the discussion more quantitatively.

4. The language should be improved throughout the manuscript.

---

## Author Comment (AC1) · 8 Jan 2018

I acknowledge the reviewers, whose reviews led to important changes and clarifications.

Reviewer 1 In the manuscript entitled "A mathematical representation of microalgae distribution in aridisol and water scarcity", Lababpour presents a mathematical model of microalgae development on surfaces. The presented model attempts to describe the interaction between cyanobacteria and soil water with the possible inclusion of climatic variables.

Although the approach of using a reaction diffusion equation of biomass and water is interesting, the manuscript does not provide in-depth tests to grasp the applicability and predictability of the model. Furthermore, the presented equations raise some questions regarding the model capability that author discusses. I enjoyed reading about the work, but have several major issues with the model and its presentation for the publication.

Response: Thank you very much for reading and for the useful comments and suggestions. The manuscript was fully revised considering the recommended suggestions. The model was developed to predict water consumption and microalgae biomass growth on the soil surfaces of aridisol as well as other photosynthetic microorganisms. The model capability is demonstrated with description of cell growth, water consumption contours, required parameters, mass transport, coupled soil biomass-water effect, and biomass distribution profiles. The simulation results of the present study show that the concentrating capability of a model can be augmented by well-selected parameter sets.

Firstly, the author describes the growth of cyanobacteria to be a function of light intensity (which should be time dependent, day and night) and water. However, the growth term is not well described in the manuscript and the solution does not reflect such behavior.

Response: We considered constant artificial continuous illumination which is common in greenhouse systems. It prohibit complexity arises of light variations. In the other word, many reports were explained the effects of natural illumination, which can be replaced by the illumination model was used in this study12. The growth term including logistic relation was explained in more detail and the biomass growth shown in Fig. 2 shows its spatial and temporal distribution.

Secondly, unlike the title highlights that the model is for aridisol, the mathematical description does not include any of the properties of such soils. Using water as one of main variables and introducing the multiplication of the porosity n might imply such application to dry soils. However, the coupled PDE only replaces the limiting nutrient to water in the conventional reaction diffusion equation for microbial growth. Especially, without including the input and output of water in the equation, the role of soil physical property (porosity in this work) cancels out, thus no soils in the equation. To be constructive, using water content instead of water concentration (it is quite difficult to understand what this term means) can be considerable to include properties of soils. The Richards equation with the saturation based soil water diffusivity can provide the proper water dynamics in dry environments.

Response: The main soil characteristics are soil texture, or particle size and porosity; and soil structure related characters, or $\theta$, h, and K. The relationship of two biomass and water state variables with soil characters are used through diffusivity and effective diffusivity. Aridisol hydraulic properties of soil moisture w, hydraulic conductivity K and soil water head, h are included in the model through model parameter of effective diffusivity coefficient, D* will be assumed known. More detail linear and non-linear diffusivity in a saturated and unsaturated soil, which is depends to factors such as tortuosity, makes the model more complex and thus omitted. The summarizing of nutrient compounds in water solution helped in simplifying the model. Otherwise it is required to write similar PDEs for all culture medium components which makes the model solution very complex. However, it is possible to expand the model equations for important nutrients such as nitrogen and phosphorous by adding similar to water PDE. The water solution was used instead of water to show inclusion of soil water nutrient ions and it was used to prohibit model complexity arises by various soil soluble ions. The effects of soil such as hydraulic properties is included in the equations through diffusivity coefficients and unit porous volume. The input and output terms, or mĬĞ = AV demonstrate the mass or volumetric flux which are depend to porosity by themselves through unit porous volume. The water concentration was changed to soil water content and the equation unit consistency was changed accordingly. Since the soil water mass flux go beyond the scope of this paper, did not discussed here and can be well explained by Richard equation. The text was modified for above suggestions accordingly.

Thirdly, the example solution of the model in this manuscript is too simple, even trivial. Simulating without any heterogeneity in the domain and uniform distributions of both variables as initial conditions should result in no differences during the dynamics (as the figures show and the author has mentioned in the result section). Furthermore, I cannot find any physical reasoning for the used boundary condition, equation (11).

Response: As a first work, even the uniform results are interesting as declare the behavior of biomass and water solution and their interrelationship. In addition, adding heterogeneity to the system of equations make their mathematical solution very difficult. We try to develop a simple model can be used for practical applications. Integrating the input and output water, in addition to diffusion, and 3-D modeling is under investigation for now. The boundary conditions were explained in more detail in the manuscript. The domain is square. In the initial condition, the domain is water saturated, without input and output water, and the biomass inoculated only in the origin of Cartesian coordination, will propagate all around in the surface as shown in Fig. 2. The biomass concentration is zero in the domain two edges, and its maximum is in the origin. For water, which is saturated in the beginning, it reduce by consumption mostly from origin, and it remains saturated in the boundaries. Two Dirichlet and two Neumann boundaries were considered.

Finally, the units provided in Table 3, do not seem to be correct. All units and dimensions need to be checked again in the model.

Response: The units were checked and corrected in the table for unit consistency.

Unfortunately, the model has major issues and needs further development. The attempt, however, to seek for a simplified representation of such a complex system is gratefully acknowledged.

Response: The comments of the reviewer were carefully considered and manuscript was revised. I hope it can be acceptable for now.

4. The language should be improved throughout the manuscript.

1. Kurano, N. & Miyachi, S. Selection of microalgal growth model for describing specific growth rate-light response using extended information criterion. J. Biosci. Bioeng. 100, 403–408 (2005). 2. Martínez, M. E., Camacho, F., Jiménez, J. M. & Espínola, J. B. Influence of light intensity on the kinetic and yield parameters of Chlorella pyrenoidosa mixotrophic growth. Process Biochem. 32, 93–98 (1997).

Please also note the supplement to this comment:
https://www.biogeosciences-discuss.net/bg-2017-359/bg-2017-359-AC1-supplement.pdf

**Supplement:**

**A mathematical representation of microalgae distribution in aridisol and water scarcity**

Abdolmajid Lababpour[1]

[1]Department of Mechanical Engineering, Shohadaye Hoveizeh University of Technology, Susangerd, 64418-78986, Iran

5    *Correspondence to*: Abdolmajid Lababpour (lababpour@shhut.ac.ir)

[revised manuscript text omitted]

$$\Omega := [0, x_L) \times [0, y_L) \times [0, z_L) \subset R^3 \tag{1}$$

in which $x_L$, $y_L$, and $z_L$ are upper and the original coordinate as lower bound in $\Omega$, respectively. The $z_L$ was assumed negligible and then horizontal 2-D was investigated. In our numerical implementation, the squared domain geometry and mesh were specified by a matrix of points, where the number of points are discretized into the 2705 nodes and 5248 triangles.

We have to choose variables that reflect the specificity of biocrust in the selected domain. From the literature we found that commonly, biomass concentration is considered to be included into the model due to its important effects on soil characteristics. In describing the biocrust development on the soil surface, it will also be logical to take into account the water role carrying nutrient for microorganisms, which is very important in further soil restoration. In addition, the water concentration on biomass growth is quite an important, and sometimes is determining factor of the restoration processes in the desert areas with water scarcity. Hence, we model the cell growth ($B$), and soil water uptake ($w$) as an interdependent mass-balance system. The temporal and spatial development of biomass and water movement variables are described by mass-balance partial differential equations. Later, new variables can be added into the model or excluded.

Any of the variables were shown with a mass balanced PDE equation in the form of diffusion-reaction equation. The variables and their interactions in each equation terms were then explained with auxiliary equations to include other factors such as light illumination. Afterward, the unit consistency in the equations were evaluated. The biocrust formation is modelled for sixty days based on the experimental data.

**Fig. 1 A schematic assumed domain geometry.**

For simplicity, a single filamentous cyanobacteria species assumed to occupy the domain in xy-plane with the *x (t)* and *y (t)*
functions in the bounded cases. A limiting water solution with concentration *w(x, y, t)* is present throughout $0 \leq x, y \leq L$. The
water containing nutrients diffuses in the horizontal plane, in a thin layer close to the surface while are consumed by cells for
growing. In the other word, when water is available, the biomass growing. Conversely, water is consumed when biomass

growing. It means we have a coupled system. Therefore, the problem requires the solution of a set of two coupled PDEs with
biomass and water availability as dependent variables. In addition, we considered the positive axis $0 \leq x, y \leq L$ in R$^3$ with a
wall placed at [x=0, x=L], [y=0, y=L], [x=L, y=L], and [x=0, y=L].

The model proposed here describes the development of biocrust with various simplification assumptions. Table 2 show the
assumptions used in the development of present biocrust model.

Table 2 Assumptions used in the BSC modelling of filamentous cyanobacteria.

For a given function *F* the general equation of second order for the biomass state variable *B(x, y, t)* is given by

$$F\left(x, y, t, B, B_{xx}, B_{yy}, B_t, \right) = 0,  \tag{2}$$

in $\Omega \in$ R$^3$. $B_{xx}$ is biomass derivative in the x, and $B_{yy}$ is the biomass derivative in the y directions. Assume that the function *F*

is sufficiently regular in its arguments. The mass-balance second order Eq. (2) for the biomass *B* is rewritten as

$$\frac{\partial B}{\partial t} = D_B\left(\frac{\partial^2 B}{\partial x^2} + \frac{\partial^2 B}{\partial y^2}\right) + \mu_B B - d_B B,  \tag{3}$$

where $D_B$ is distribution coefficient of biomass in the x and y direction, $\mu$ is specific growth rate, and *d* is cell death coefficient.
The soil biomass distribution depends on the cell growth as well as soil physical characteristics. The dependency to cell growth
and death is shown by (μ-d) coefficients, while the dependency on the soil texture and soil water content are demonstrated by
$D_B$ coefficient.

To include soil factors especially for drylands, we focused on effective diffusion coefficient per retardation factor, $D^*/R_d$.
Effective diffusion coefficient is less than diffusivity, *D,* in solution which arises of more tortuous pathway in soil and less
mass flux. To include soil porosity, etc. Eq. (3) becomes as

$$nV\frac{\partial B}{\partial t} = nVD_B\left(\frac{\partial^2 B}{\partial x^2} + \frac{\partial^2 B}{\partial y^2}\right) + nV\mu_B B - nVd_B B,  \tag{4}$$

where *n* is porosity and *V* is volume.

The effect of light illumination, soil water content and maximum biomass, $B_{max}$ on specific growth rate, $\mu_B$ are defined by

empirical relationship has been obtained from a study on the photosynthesis of phytoplankton in the sea (DiToro et al., 1971),
Mickaelis-Menton, and the logistic equation.

$$\mu(t) = \mu_{max}.f(I).g(B_{max}).h(w)$$

$$\mu(I) = \mu_{max}.\frac{I_{av}}{I_s}e^{1-\frac{I_{av}}{I_s}}, \qquad\qquad (5)$$

$$\mu(B_{max}) = \mu_{max}(1 - \frac{B}{B_{max}})$$

$$\mu(w) = \mu_{max}(\frac{w}{w + k_w})$$

where $I_{av}$ the average or instantaneous illumination rate, and $I_s$ are the optimal illumination rate ($\mu$mol photon/m$^2\cdot$d). The logistic model was selected for cell growth, where $\mu_{max}$ the maximum growth rate (d$^{-1}$), and $B_{max}$ are maximum biomass concentration (g/l) (Chambon et al., 2013). The relationship of water solution consumption and growth rates were shown via two parameters of maximum growth rate $\mu_{max}$ and the yield coefficient with respect to the water solution. The link between yield coefficient and water solution utilization is represented linearly which relate the yield coefficient to the specific growth rate of biomass (Kovárová-Kovar and Egli, 1998).

The inclusion of illumination term in the Eq. (4) implies that the model is applicable for the microorganisms with photosynthetic activities. Biomass production increases as a function of light intensity until an optimal intensity is reached, and beyond that optimal value, production varies in accordance with the type of light source. In this study, continuous lighting with constant irradiation was applied, can be supported by artificial lamps such as fluorescent.

The mass-balance equation for saturated soil water in a horizontal top soil layer (biocrust) can be described by the following equation

$$nV\frac{\partial w}{\partial t} = nVD_w\left(\frac{\partial^2 w}{\partial x^2} + \frac{\partial^2 w}{\partial y^2}\right) + R(t) + I(w(t)) - ET(w(t)) - LQ(w(t)) - nV\frac{1}{Y_{B/w}}(\mu_B - d_B)B \qquad (6)$$

The terms in this equation from the left are for temporal $nV\,\partial w/\partial t$ and spatial $nVD_w\,(\partial^2 w/\partial\,x^2+\partial\,^2 w/\partial y^2)$ soil water profile, time $t$, soil water content $nV$ ($A$ surface area, $L$ length and $n$ porosity), water content $w$, rainfall $R(t)$(kg/d), irrigation $I(w(t))$ (kg/d), evapotranspiration $ET(w(t))$(kg/d), the combination of percolation and runoff water losses $LQ(w(t))$(kg/d), and water uptake by photosynthetic microorganisms $nV\mu_B B/Y_{B/w}$ (kg/d), in which $\mu_B$ and $Y_{B/w}$ are for specific growth rate and conversion yield (-), respectively (Albano et al., 2017). The term rainfall and irrigation are modelled according to (Bartlett et al., 2015), the actual evapotranspiration is estimated for arid and semiarid land areas by empirical data and the relationship described as $Et_a + ET_p = 2ET_w$, in which $Et_a$ is actual evapotranspiration, $ET_p$ is climatic parameter, and $ET_w$ is wet environmental evapotranspiration, soil water losses by percolation and runoff are modelled according to (Bartlett et al., 2015).

The soil water consumption by photosynthetic microorganisms are represented by logistic equation multiplied by yield conversion coefficient, $Y_{B/w}$ (Albano et al., 2017; Baudena et al., 2013).

From continuity equation and with no water input and output assumption from rain, irrigation, percolation and evapotranspiration, Eq. (6) reduce to Eq. (7) as

$$nV \frac{\partial W}{\partial t} = nVD_w \left( \frac{\partial^2 W}{\partial x^2} + \frac{\partial^2 W}{\partial y^2} \right) - nV \frac{1}{Y_{B/w}} (\mu_B - d_B)B \qquad (7)$$

may be extended to include soil water output and inputs by adding mass flow rates. The soil water content per unit volume of soil porosity was considered as $w = \partial W / n \partial V_{por}$ be the intensive value. We can assuming homogeneous bed, and then Eq. (7) reduces to

$$\frac{\partial w}{\partial t} = D_w \left( \frac{\partial^2 w}{\partial x^2} + \frac{\partial^2 w}{\partial y^2} \right) - \frac{1}{Y_{b/w}} (\mu_b - d_b)b \qquad (8)$$

The term $D^*/ R_d$ was determined.

$$D^*/Rd = D_0 . \tau / Rd \qquad (7)$$

in which $\tau$ is for tortuosity factors (Shackelford and Daniel, 1991).

This equation shows temporal and spatial soil water in the x and y directions because of soil water diffusion, and its consumption by microorganisms. Various soil types porosity of arid areas can be included in the model through parameter $n$, which is commonly in the range of $0.3 - 075$ (Ci and Yang, 2009). It is also empirical relations are available may be used for converting soil porosity to other soil physical characteristics such as particle size and density (Foster and Miklavcic, 2013).

At time zero, water solution is at its saturated state and uniformly distributed in the space $w_{x,y}^{t=0} = p$ for all *(x, y, 0)* elements. The initial conditions associated with the biomass equation is defined as

$$B(x, y, t = 0) = B_0(x, y) = 0.03, \qquad x \in \Omega, \qquad (9)$$

$$w(x, y, t = 0) = w_0(x, y) = p, \qquad x \in \Omega, \qquad (10)$$

where $B_0$ and $w_0$ are defined in all the plate at t = 0. Based on the experimental data, the initial biomass concentration of 0.03 kg/m³ and soil water content in the range of $0.1 < p < 0.8$ were considered as initial conditions. The saturated water fraction is influenced by soil physical parameters such as porosity. This show that all of domain points have similar conditions of biomass and water content in the beginning.

The boundary conditions for the system shown by Eqs. (4), and (7) are given by

$$B(x_{L=0}, y, t) = \frac{\partial B}{\partial t} \qquad x \in \Omega, \qquad y \in \Omega, \qquad t \geq 0$$

$$B(x_L, y, t) = 0 \qquad x \in \Omega, \qquad y \in \Omega, \qquad t \geq 0$$

$$B(x, y_{L=0}, t) = \frac{\partial B}{\partial t} \qquad x \in \Omega, \qquad y \in \Omega, \qquad t \geq 0$$

$$B(x, y_L, t) = 0 \qquad x \in \Omega, \qquad y \in \Omega, \qquad t \geq 0$$

(11)

where *B (x, y, t)* and *w (x, y, t)* are given as homogeneous Dirichlet bounded boundary conditions is imposed between horizontal coordinates $0 \leq x < x_L$ and $0 \leq y < y_L$. In our case, we supposed $x_L = y_L = 2$ m. Other two boundary conditions follow Neumann formula.

The nutrient uptake from water solution is governed by a set of specific boundary conditions. For photosynthetic microorganisms expanding in the horizontal domain, the water content is maximal and constant outside the plane. With the neglecting mass flow inside the domain, then the boundary conditions for water solution are given by Dirichlet and Neumann formula

$$w(x_{L=0}, y, t) = \frac{\partial B}{\partial t} \quad x \in \Omega, y \in \Omega, t \geq 0$$

$$w(x_L, y, t) = 0.55 \quad x \in \Omega, y \in \Omega, t \geq 0 \ (for \ aridisol)$$

$$w(x, y_{L=0}, t) = \frac{\partial B}{\partial t} \quad x \in \Omega, y \in \Omega, t \geq 0$$

$$w(x, y_L, t) = 0.55 \quad x \in \Omega, y \in \Omega, t \geq 0 \ (for \ aridisol)$$

$p$ is fraction of total available soil water, which was assumed in the range of $0.1 < p < 0.8$ for various soil types.

Gathering the above PDE equations, we have the model incorporated two 2-D PDE equations, 2 initial conditions, 8 boundary conditions, and 8 parameters divided to biomass and soil water solution related variables in a same domain. The involved model parameters estimated from experimental data or literature are summarized in Table 3.

Table 3 Summary of parameters used for the computation of model and their characteristics.

**2.2 Numerical model solution algorithm, codes and software**

The numerical solution of nonlinear system of coupled partial differential equations represented by Eqs. (4), and (7) were provided using the Matlab software version R2014a (8.3.0.532) (The Mathwork, Inc., USA). We used the parabolic solver for PDEs solution. In the numerical scheme, the time derivative with respect to $t$ and space partial derivative with respect to $x$ and $y$ in Eqs. (4), and (7) are discretized to a consistent order of approximation using central difference. Equations (4) and (7) were written in matrix coefficient form and solved to find the biomass and water solution horizontal profile in each time and space point on the soil surface. For each time step, the specific growth rate, $\mu_B$, was calculated from light function $f(I)$, and soil water function $f(w)$, in each element from the previous time space. These specific growth rate values then used to calculate biomass and soil water content in the current time step. The solver uses finite element method (FEM) for solving partial differential equations (Boltz et al., 2010). We split the program into three part functions. The suit of Matlab script was used to generate domain geometry and meshing for xy discretization, tlist for time discretization, describe PDEs parameter and function coefficients, solver setting, and visualization and results presentation. The initial and boundary conditions were prepared in a separate function files.

**3 Results**

**3.1 Total biomass**

The developed mathematical model is able to predict concentration profiles for biomass, soil water as substrate, and light profiles in the biocrust as well as carbon dioxide concentrations and solute nutrient. Figure 2 shows a typical time course of a single simulation biomass profile for the biocrust. According to Fig. 2, the cyanobacteria biomass concentration increase as time pass during cultivation period close to the soil surface, which is in accordance with experimental observations in the petri dish cultures (Lababpour and Kaviani, 2016). Although the spatial distribution of biomass is not homogeneous in real soil conditions, as it is function of variables such as soil non-homogeneous structure and behaviour of microorganisms, the model show uniform biomass distribution in constructed biocrust as expected. A relatively homogeneous distribution on the surface can be observed in experimental biocrust set up. This difference is shown to be practically are not significant. However, the approximately close biomass production appears by the model is an expectable observation under the conditions of simulation. Besides the uncertainty caused by additional model parameters used in the biocrust model, there also be an added uncertainty caused by the model structure and simplification assumptions is not considered in this study.

The increasing biomass within the biocrust stems from a reaction-diffusion interaction. The model predicts the biomass concentration gradient as well as the soil water content as a result of suction gradient is established by biomass water uptake as the cell began to growth. This water gradient forces the water to move towards domain's centre from the boundaries. This process continuous as long as the domain's soil has enough water storage. Biomass concentration is increasing near the biocrust interface and becomes reduced in the biocrust interior. However its distribution is homogeneous in the domain x and y directions surface. The profile of viable cell fraction is reversed in which cell numbers is in higher numbers of the biocrust surface and are decrease effectively near the domain boundaries. Viable cell numbers and biocrust thickness reduction were dissimilar measures of light efficacy. The general behaviour predicted by the model is in good qualitative agreement with an experimental observations. As the initial seeding was considered similar in domain, therefore, biomass distribution would be homogeneous in all of the surface.

Fig. 2 The solution of biomass partial differential equation with two Dirichlet and two Neumann boundaries (and initial condition of 0.03). Results of the temporal and spatial biomass concentration distribution profile in the x and y surface. (a) Temporal distribution, and (b) spatial distribution with surface plot.

The modelling of soil water is more complex as it is not only dependent to up taken by microorganisms, but also dependent to forces acting on the soil water mass balance such as soil porespaces, evapotranspiration, and input-output water flow to soil. If we assume no input-output water flow, and water is free for movement in the soil, the model prediction reveals of a logarithmic water reduction curve during cultivation period. The decreasing speed in this case is most inversely corresponds to cell growth rate. Figure 3 reveals time profile of soil water in the domain during cultivation period in the horizontal surface.

٢٢٥ Fig. 3 The solution of diffusion partial differential equation of soil water content with two Dirichlet and two Neumann boundaries (and initial condition of 0.55). Results of the soil water content 
[revised manuscript text omitted]

Shackelford, C. D. and Daniel, D. E.: Diffusion in saturated soil. I: Background, J. Geotech. Eng., 117(3), 467–484, 1991.

Systems, T.: Systems and control group identification of algae growth kinetics, Wageningen. [online] Available from:

http://edepot.wur.nl/9247, 2009.

Vasilyeva, N. A., Ingtem, J. G. and Silaev, D. A.: Nonlinear dynamical model of microorganism growth in soil, Comput. Math. Model., 27(2), 172–180, doi:10.1007/s10598-016-9312-7, 2016.

Vidriales-Escobar, G., Rentería-Tamayo, R., Alatriste-Mondragón, F. and González-Ortega, O.: Mathematical modeling of a composting process in a small-scale tubular bioreactor, Chem. Eng. Res. Des., 120, 360–371, doi:10.1016/j.cherd.2017.02.006, 2017.

٣٥٥

Table 1

| BSC | Biofilm |
|---|---|
| Has higher species diversity | Has single or few species diversity |
| The associated communities are bacteria, fungi, lichen and mosses | The associated communities are similar microorganisms e.g. bacteria |
| In addition to cyanobacteria, other microorganisms such as fungi participate in EPS production. | Cyanobacteria are main EPS producers |
| Coral and fine particles are main components | Only biofilm components are constituent species |
| They are more porous | They are less porous |
| *Microcoleus* lack UV protecting pigments, then stratifies in the sub-surface, while some other such as *Nostoc* having UV screening pigments stratify at the soil surface. | Relatively homogeneous |
| The substratum cannot be defined | The substratum is fixed flat plate |
| There is no detachments, and the decayed cell remain in the soil. | The cell detached from the biofilm surface. |

Table 2

| Assumed condition | Real condition |
| --- | --- |
| Two-dimensional (2D) - BSC structure parallel to soil surface | Three-dimensional (3D) biocrust structure |
| Homogenous biofilm structure, (uniformly thick planar aggregate) | Heterogeneous biocrust structure (physical, chemical, biological and geometrical) |
| Biocrust is solid matrix and rigid structure | Biocrust is multiphase system |
| Pure species biocrust | Diverse multispecies biocrust |
| Fixed biocrust boundary conditions | Moving biocrust boundary conditions |
| Biocrust is flat | Biocrust is fractal shape |
| Biofilm thickness is constant. | Biocrust volume and thickness is variable |
| Diffusion rate of substrate is constant | Diffusion rate of substrate is variable |
| Temperature is constant and in optimum conditions | There is temperature fluctuation in the biocrust |
| All variables over areas parallel to the substratum are constant. | Variables are not uniform over surface area parallel to substratum |
| Light kinetics describe by DiToro et al kinetics | Kinetic of light variation is complex inside of the biocrust |
| Substrate concentration and environmental parameters are not limiting except light illumination and water | Many substrate and environmental parameters are limiting and complex inside of the biocrust |
| Light enters the biocrust domain perpendicular to surface | Light angle and intensity change with time and locations |
| The domain boundaries are no diffusible and fix | The biocrust boundaries are not fix and growing fractal. |

Table 3

| Name | Symbol | Unit | Value | Ref. |
|------|--------|------|-------|------|
| Biomass distribution coefficient | $D_B$ | $m^2/d$ | 6.25E-4 | a |
| Maximum specific growth rate | $\mu_{max}$ | $d^{-1}$ | 0.4 | b |
| Water distribution coefficient | $D_w$ | $m^2/d$ | 6.25E-2 | a |
| Effective water diffusion coefficient | $D_w^*$ | $m^2/d$ | | d |
| Retardation factor | $R_d$ | - | | d |
| Optimal light illumination coefficient | $I_s$ | $\mu$mol photone/$m^2\cdot d$ | 0.007 | a |
| Michelis – Menton coefficient | $K_w$ | W/W | 0.01 | b |
| Soil porosity $(V_V/V_T)^*$ | n | % | 0.3-0.7 | c |
| Soil water content (mass percent = $m_{biomass}/m_{soil \times 100 \ (W/W)}$) | V | W/W | 0.2 | This study |
| Time | t | d | 60 | This study |

a, (DiToro et al., 1971); b, (Systems, 2009); c,(Schiavone, 2016); d, (Shackelford and Daniel, 1991).

$V_V$ is the volume of void-space (such as fluids) and $V_T$ is the total or bulk volume of material, including the solid and void components.

Fig. 1

[Figure]

**a**

[Figure]

Fig. 3

**a**

[Figure]

[Figure]

---

## Author Comment (AC2) · 8 Jan 2018

I acknowledge the reviewers, whose reviews led to important changes and clarifications.

Reviewer 2 Report on the paper "A mathematical representation of microalgae distribution in aridisol and water scarcity" by A. Lababpour November 6, 2017

The author presents a model consisting of two diffusion-reaction equations for biomass and water development. The resulting model equations are solved using a Matlab toolbox. The author thoroughly discusses extensions of the modeling approach and

outline directions of further research. He particularly discusses influencing factors for the three dimensional development of the soil biocrust. Although this part is quite well explained and the topic itself is of interest, I do not recommend the manuscript in its current stage for publishing in Biogeosciences. For details see the following comments below.

Response The resulting model equations are solved using parabolic solver in Matlab 2014a (The Mathwork, Inc.) (Not by Matlab toolbox). The program was split into geometry definition, initial condition, boundary conditions and solver for both PDEs. The biocrust system is very complex, and the proposed model is preliminary trial compare to real conditions. Model could simply integrate illumination, biomass growth, soil water, and soil physical properties. These parameters are important in recognitions of soil restoration systems.

1. The model is very simplified although possible reasonable extensions are discussed. The predictive power is therefore questionable. A quantitative discussion e.g. in comparison with with data should be added.

Response To improve the predictive power of model, a more data and discussion were included.

2. The model under consideration and its precise mathematical formulation remains unclear. The author should clearly differentiate what could be done and what is done within the current research. The model that is implemented it not clearly stated in its whole at any point. Exactly this model should be summarized once in the paper including the domain of definition, the used variables and parameters.

Response Some parts of text were revised to clarify the model subsections such as domain, state variables and parameters. The goal of this research is to develop a simple model to predict the growth of photosynthetic microorganisms and their interactions with soil water content in water shortage conditions of arid areas. The results of this research provide some insight of such interrelationships.

(a) The geometric setting is unclear. First the three dimensional domain [0, Lx] × [0, Ly] × [0, Lz] is discussed. This is thereafter changed to the half space (which actually is not the half space but the first quadrant). Thereafter a square [0, 4]2 is considered.

Response The geometric setting was revised in the text to show more accurate the physical conditions. The selected domain was explained as "The u (x, y, t) satisfies in $\Omega$ and then, domain may be represented as $\Omega$âĹ̌ű=[0,x_L]×[0,y_L]×[0,y_L)âŁĆRˆ3 (1) in which xL, yL, and Lz are upper and the original coordinate as lower bound in $\Omega$, respectively. The zL was assumed negligible and then horizontal 2-D was investigated. ". Half space changed to positive semi-axis. "a square [0, 4]2 was removed.

(b) It is unclear how the porosity enters the investigations. The equation (4) is reasonable only in case that the porosity is independent of space. Otherwise I would expect that the diffusion terms reads 1âĹŊĞãČż (nV DBâĹŊĞB). However, it seems that there is no modulation of the porosity throughout the simulation scenario, e.g. in the initial conditions.

Response Yes the porosity is independent of space and was considered uniform throughout the selected small domain. The porosity was included in simulation by D* constant parameter, mentioned in Table 2. The porosity was considered as one of critical parameters of soil can affect main both state variables of biomass and soil water content and can be related to other soil parameters such as particle size, etc. we assumed homogeneous soil texture. And therefore porosity is constant. It is common to apply porosity as a constant for modeling. Porosity is considered in all modeling steps including initial conditions. The porosity inclusion in the model was modified in the model formulation. The equations presented in this paper do not consider the effect of coupled flow processes, i.e., solute transport due to hydraulic, thermal, or electrical gradients.

(c) The inclusion of the illumination rate into the growth term of equation (4) is unclear. How is I related to $\mu$ and how is it determined/defined in the simulation scenario? Along

the same lines d is undefined in (8). Does the logistic model enter the proposed model in terms the growth rate $\mu$ in (4) and (8) and if so how does it?

Response The inclusion of illumination and logistic model was revised in the text as suggested. The function $\mu$ = f (I).f (B) indicate the situation of illumination and growth in equation (4). The coefficient of I was selected from literature such as Lababpour (2004), and their effects was included through simulation by $\mu$. Dw is water diffusion which is included in the text. dB is biomass death rate represented highlighted in line 111.

(d) The statement of boundary condition in (11) makes no sense. First the boundary conditions must be prescribed on the boundary of the domain rather that in the domain itself. Second the variables are independent quantities. I assume the author means B = 0.02 and w = 0.2 are prescribed on the boundary.

Response To adopt reviewer comments, the position of boundary conditions were changed. Accordingly the results was modified. Both Dirichlet and Neumann boundary conditions were selected for simulation of variable's profile. The biomass concentration is highest in the origin and is equal to initial biomass concentration and zero in the domain's boundaries. The amount of biomass remain zero in the boundaries constant during simulation. The soil water content is saturated soil water throughout the domain in the beginning of simulation and remain saturated, constant in the boundaries. In the other word, the soil water consumption is highest in the origin and zero in the boundaries. Modifications was performed in the text accordingly. Actually it is not easy to describe soil water content, as it has several state terms such as saturated, welting, field capacity... Here we considered the range of saturated and wilting point between 0.3-0.7.

3. The outcome is of the simulations are not convincing: (a) The scaling in Figure 2 and 3 is unclear. Variations are seen in order $10-4$ or even below. This seems not to be relevant compared to the initial/boundary values or even the reference value of 0.7921

and 0.5198 in Figure 2 and 3. The smallness of the variations could even be due to numerical or rounding errors. It is remarkable that the values for B and w are within the same order of magnitude directly although starting with one order of magnitude difference initially.

Response The figures 2 and 3 were modified accordingly as suggested by reviewer.

(b) The initial conditions are not matched for t = 0 in Figure 2 and 3. This does not make the results reliable.

Response The initial conditions for biomass is 0.03 kg.m-3 of soil porous volume at t = 0, and increased to 0.26 during 60 days growth period. The figures 2 and 3 were modified and corrected.

(c) It is unclear for which time the spatial distribution is plotted. The values are very small (order of $10-4$) compared to the chosen initial values. Is zero maybe a steady state solution?

Response The 60 days was selected for simulation based on the normal experimental growth period of selected microalga. The time distribution in the figures 2 and 3 were modified as suggested. Unsteady state cell growth was applied throughout simulation.

(d) Explicit comparison to data is missing. This could maybe shed some light into the results and make the discussion more quantitatively.

Response The text was modified as suggested.

4. The language should be improved throughout the manuscript. 1. Kurano, N. & Miyachi, S. Selection of microalgal growth model for describing specific growth rate-light response using extended information criterion. J. Biosci. Bioeng. 100, 403–408 (2005). 2. Martínez, M. E., Camacho, F., Jiménez, J. M. & Espínola, J. B. Influence of light intensity on the kinetic and yield parameters of Chlorella pyrenoidosa mixotrophic growth. Process Biochem. 32, 93–98 (1997).
Please also note the supplement to this comment:
https://www.biogeosciences-discuss.net/bg-2017-359/bg-2017-359-AC2-supplement.pdf

─────────────────────────

**Supplement:**

**A mathematical representation of microalgae distribution in aridisol and water scarcity**

Abdolmajid Lababpour[1]

[1]Department of Mechanical Engineering, Shohadaye Hoveizeh University of Technology, Susangerd, 64418-78986, Iran

5    *Correspondence to*: Abdolmajid Lababpour (lababpour@shhut.ac.ir)

[revised manuscript text omitted]

$$\Omega := [0, x_L) \times [0, y_L) \times [0, z_L) \subset R^3 \tag{1}$$

in which $x_L$, $y_L$, and $z_L$ are upper and the original coordinate as lower bound in $\Omega$, respectively. The $z_L$ was assumed negligible and then horizontal 2-D was investigated. In our numerical implementation, the squared domain geometry and mesh were specified by a matrix of points, where the number of points are discretized into the 2705 nodes and 5248 triangles.

We have to choose variables that reflect the specificity of biocrust in the selected domain. From the literature we found that commonly, biomass concentration is considered to be included into the model due to its important effects on soil characteristics. In describing the biocrust development on the soil surface, it will also be logical to take into account the water role carrying nutrient for microorganisms, which is very important in further soil restoration. In addition, the water concentration on biomass growth is quite an important, and sometimes is determining factor of the restoration processes in the desert areas with water scarcity. Hence, we model the cell growth ($B$), and soil water uptake ($w$) as an interdependent mass-balance system. The temporal and spatial development of biomass and water movement variables are described by mass-balance partial differential equations. Later, new variables can be added into the model or excluded.

Any of the variables were shown with a mass balanced PDE equation in the form of diffusion-reaction equation. The variables and their interactions in each equation terms were then explained with auxiliary equations to include other factors such as light illumination. Afterward, the unit consistency in the equations were evaluated. The biocrust formation is modelled for sixty days based on the experimental data.

**Fig. 1 A schematic assumed domain geometry.**

For simplicity, a single filamentous cyanobacteria species assumed to occupy the domain in xy-plane with the *x (t)* and *y (t)*
functions in the bounded cases. A limiting water solution with concentration *w(x, y, t)* is present throughout $0 \leq x, y \leq L$. The
water containing nutrients diffuses in the horizontal plane, in a thin layer close to the surface while are consumed by cells for
growing. In the other word, when water is available, the biomass growing. Conversely, water is consumed when biomass

growing. It means we have a coupled system. Therefore, the problem requires the solution of a set of two coupled PDEs with
biomass and water availability as dependent variables. In addition, we considered the positive axis $0 \leq x, y \leq L$ in R$^3$ with a
wall placed at [x=0, x=L], [y=0, y=L], [x=L, y=L], and [x=0, y=L].

The model proposed here describes the development of biocrust with various simplification assumptions. Table 2 show the
assumptions used in the development of present biocrust model.

Table 2 Assumptions used in the BSC modelling of filamentous cyanobacteria.

For a given function *F* the general equation of second order for the biomass state variable *B(x, y, t)* is given by

$$F\left(x, y, t, B, B_{xx}, B_{yy}, B_t, \right) = 0,  \tag{2}$$

in $\Omega \in$ R$^3$. $B_{xx}$ is biomass derivative in the x, and $B_{yy}$ is the biomass derivative in the y directions. Assume that the function *F*

is sufficiently regular in its arguments. The mass-balance second order Eq. (2) for the biomass *B* is rewritten as

$$\frac{\partial B}{\partial t} = D_B\left(\frac{\partial^2 B}{\partial x^2} + \frac{\partial^2 B}{\partial y^2}\right) + \mu_B B - d_B B,  \tag{3}$$

where $D_B$ is distribution coefficient of biomass in the x and y direction, $\mu$ is specific growth rate, and *d* is cell death coefficient.
The soil biomass distribution depends on the cell growth as well as soil physical characteristics. The dependency to cell growth
and death is shown by (μ-d) coefficients, while the dependency on the soil texture and soil water content are demonstrated by
$D_B$ coefficient.

To include soil factors especially for drylands, we focused on effective diffusion coefficient per retardation factor, $D^*/R_d$.
Effective diffusion coefficient is less than diffusivity, *D,* in solution which arises of more tortuous pathway in soil and less
mass flux. To include soil porosity, etc. Eq. (3) becomes as

$$nV\frac{\partial B}{\partial t} = nVD_B\left(\frac{\partial^2 B}{\partial x^2} + \frac{\partial^2 B}{\partial y^2}\right) + nV\mu_B B - nVd_B B,  \tag{4}$$

where *n* is porosity and *V* is volume.

The effect of light illumination, soil water content and maximum biomass, $B_{max}$ on specific growth rate, $\mu_B$ are defined by

empirical relationship has been obtained from a study on the photosynthesis of phytoplankton in the sea (DiToro et al., 1971),
Mickaelis-Menton, and the logistic equation.

$$\mu(t) = \mu_{max}.f(I).g(B_{max}).h(w)$$

$$\mu(I) = \mu_{max}.\frac{I_{av}}{I_s}e^{1-\frac{I_{av}}{I_s}}, \qquad\qquad (5)$$

$$\mu(B_{max}) = \mu_{max}(1 - \frac{B}{B_{max}})$$

$$\mu(w) = \mu_{max}(\frac{w}{w + k_w})$$

where $I_{av}$ the average or instantaneous illumination rate, and $I_s$ are the optimal illumination rate ($\mu$mol photon/m$^2\cdot$d). The logistic model was selected for cell growth, where $\mu_{max}$ the maximum growth rate (d$^{-1}$), and $B_{max}$ are maximum biomass concentration (g/l) (Chambon et al., 2013). The relationship of water solution consumption and growth rates were shown via two parameters of maximum growth rate $\mu_{max}$ and the yield coefficient with respect to the water solution. The link between yield coefficient and water solution utilization is represented linearly which relate the yield coefficient to the specific growth rate of biomass (Kovárová-Kovar and Egli, 1998).

The inclusion of illumination term in the Eq. (4) implies that the model is applicable for the microorganisms with photosynthetic activities. Biomass production increases as a function of light intensity until an optimal intensity is reached, and beyond that optimal value, production varies in accordance with the type of light source. In this study, continuous lighting with constant irradiation was applied, can be supported by artificial lamps such as fluorescent.

The mass-balance equation for saturated soil water in a horizontal top soil layer (biocrust) can be described by the following equation

$$nV\frac{\partial w}{\partial t} = nVD_w\left(\frac{\partial^2 w}{\partial x^2} + \frac{\partial^2 w}{\partial y^2}\right) + R(t) + I(w(t)) - ET(w(t)) - LQ(w(t)) - nV\frac{1}{Y_{B/w}}(\mu_B - d_B)B \qquad (6)$$

The terms in this equation from the left are for temporal $nV\,\partial w/\partial t$ and spatial $nVD_w\,(\partial^2 w/\partial\,x^2+\partial\,^2 w/\partial y^2)$ soil water profile, time $t$, soil water content $nV$ ($A$ surface area, $L$ length and $n$ porosity), water content $w$, rainfall $R(t)$(kg/d), irrigation $I(w(t))$ (kg/d), evapotranspiration $ET(w(t))$(kg/d), the combination of percolation and runoff water losses $LQ(w(t))$(kg/d), and water uptake by photosynthetic microorganisms $nV\mu_B B/Y_{B/w}$ (kg/d), in which $\mu_B$ and $Y_{B/w}$ are for specific growth rate and conversion yield (-), respectively (Albano et al., 2017). The term rainfall and irrigation are modelled according to (Bartlett et al., 2015), the actual evapotranspiration is estimated for arid and semiarid land areas by empirical data and the relationship described as $Et_a + ET_p = 2ET_w$, in which $Et_a$ is actual evapotranspiration, $ET_p$ is climatic parameter, and $ET_w$ is wet environmental evapotranspiration, soil water losses by percolation and runoff are modelled according to (Bartlett et al., 2015).

The soil water consumption by photosynthetic microorganisms are represented by logistic equation multiplied by yield conversion coefficient, $Y_{B/w}$ (Albano et al., 2017; Baudena et al., 2013).

From continuity equation and with no water input and output assumption from rain, irrigation, percolation and evapotranspiration, Eq. (6) reduce to Eq. (7) as

$$nV \frac{\partial W}{\partial t} = nVD_w \left( \frac{\partial^2 W}{\partial x^2} + \frac{\partial^2 W}{\partial y^2} \right) - nV \frac{1}{Y_{B/w}} (\mu_B - d_B)B \qquad (7)$$

may be extended to include soil water output and inputs by adding mass flow rates. The soil water content per unit volume of soil porosity was considered as $w = \partial W / n \partial V_{por}$ be the intensive value. We can assuming homogeneous bed, and then Eq. (7) reduces to

$$\frac{\partial w}{\partial t} = D_w \left( \frac{\partial^2 w}{\partial x^2} + \frac{\partial^2 w}{\partial y^2} \right) - \frac{1}{Y_{b/w}} (\mu_b - d_b)b \qquad (8)$$

The term $D^*/ R_d$ was determined.

$$D^*/Rd = D_0 . \tau / Rd \qquad (7)$$

in which $\tau$ is for tortuosity factors (Shackelford and Daniel, 1991).

This equation shows temporal and spatial soil water in the x and y directions because of soil water diffusion, and its consumption by microorganisms. Various soil types porosity of arid areas can be included in the model through parameter $n$, which is commonly in the range of $0.3 - 075$ (Ci and Yang, 2009). It is also empirical relations are available may be used for converting soil porosity to other soil physical characteristics such as particle size and density (Foster and Miklavcic, 2013).

At time zero, water solution is at its saturated state and uniformly distributed in the space $w_{x,y}^{t=0} = p$ for all *(x, y, 0)* elements. The initial conditions associated with the biomass equation is defined as

$$B(x, y, t = 0) = B_0(x, y) = 0.03, \qquad x \in \Omega, \qquad (9)$$

$$w(x, y, t = 0) = w_0(x, y) = p, \qquad x \in \Omega, \qquad (10)$$

where $B_0$ and $w_0$ are defined in all the plate at t = 0. Based on the experimental data, the initial biomass concentration of 0.03 kg/m³ and soil water content in the range of $0.1 < p < 0.8$ were considered as initial conditions. The saturated water fraction is influenced by soil physical parameters such as porosity. This show that all of domain points have similar conditions of biomass and water content in the beginning.

The boundary conditions for the system shown by Eqs. (4), and (7) are given by

$$B(x_{L=0}, y, t) = \frac{\partial B}{\partial t} \qquad x \in \Omega, \qquad y \in \Omega, \qquad t \geq 0$$

$$B(x_L, y, t) = 0 \qquad x \in \Omega, \qquad y \in \Omega, \qquad t \geq 0$$

$$B(x, y_{L=0}, t) = \frac{\partial B}{\partial t} \qquad x \in \Omega, \qquad y \in \Omega, \qquad t \geq 0$$

$$B(x, y_L, t) = 0 \qquad x \in \Omega, \qquad y \in \Omega, \qquad t \geq 0$$

(11)

where *B (x, y, t)* and *w (x, y, t)* are given as homogeneous Dirichlet bounded boundary conditions is imposed between horizontal coordinates $0 \leq x < x_L$ and $0 \leq y < y_L$. In our case, we supposed $x_L = y_L = 2$ m. Other two boundary conditions follow Neumann formula.

The nutrient uptake from water solution is governed by a set of specific boundary conditions. For photosynthetic microorganisms expanding in the horizontal domain, the water content is maximal and constant outside the plane. With the neglecting mass flow inside the domain, then the boundary conditions for water solution are given by Dirichlet and Neumann formula

$$w(x_{L=0}, y, t) = \frac{\partial B}{\partial t} \quad x \in \Omega, y \in \Omega, t \geq 0$$

$$w(x_L, y, t) = 0.55 \quad x \in \Omega, y \in \Omega, t \geq 0 \ (for \ aridisol)$$

$$w(x, y_{L=0}, t) = \frac{\partial B}{\partial t} \quad x \in \Omega, y \in \Omega, t \geq 0$$

$$w(x, y_L, t) = 0.55 \quad x \in \Omega, y \in \Omega, t \geq 0 \ (for \ aridisol)$$

$p$ is fraction of total available soil water, which was assumed in the range of $0.1 < p < 0.8$ for various soil types.

Gathering the above PDE equations, we have the model incorporated two 2-D PDE equations, 2 initial conditions, 8 boundary conditions, and 8 parameters divided to biomass and soil water solution related variables in a same domain. The involved model parameters estimated from experimental data or literature are summarized in Table 3.

Table 3 Summary of parameters used for the computation of model and their characteristics.

**2.2 Numerical model solution algorithm, codes and software**

The numerical solution of nonlinear system of coupled partial differential equations represented by Eqs. (4), and (7) were provided using the Matlab software version R2014a (8.3.0.532) (The Mathwork, Inc., USA). We used the parabolic solver for PDEs solution. In the numerical scheme, the time derivative with respect to $t$ and space partial derivative with respect to $x$ and $y$ in Eqs. (4), and (7) are discretized to a consistent order of approximation using central difference. Equations (4) and (7) were written in matrix coefficient form and solved to find the biomass and water solution horizontal profile in each time and space point on the soil surface. For each time step, the specific growth rate, $\mu_B$, was calculated from light function $f(I)$, and soil water function $f(w)$, in each element from the previous time space. These specific growth rate values then used to calculate biomass and soil water content in the current time step. The solver uses finite element method (FEM) for solving partial differential equations (Boltz et al., 2010). We split the program into three part functions. The suit of Matlab script was used to generate domain geometry and meshing for xy discretization, tlist for time discretization, describe PDEs parameter and function coefficients, solver setting, and visualization and results presentation. The initial and boundary conditions were prepared in a separate function files.

**3 Results**

**3.1 Total biomass**

The developed mathematical model is able to predict concentration profiles for biomass, soil water as substrate, and light profiles in the biocrust as well as carbon dioxide concentrations and solute nutrient. Figure 2 shows a typical time course of a single simulation biomass profile for the biocrust. According to Fig. 2, the cyanobacteria biomass concentration increase as time pass during cultivation period close to the soil surface, which is in accordance with experimental observations in the petri dish cultures (Lababpour and Kaviani, 2016). Although the spatial distribution of biomass is not homogeneous in real soil conditions, as it is function of variables such as soil non-homogeneous structure and behaviour of microorganisms, the model show uniform biomass distribution in constructed biocrust as expected. A relatively homogeneous distribution on the surface can be observed in experimental biocrust set up. This difference is shown to be practically are not significant. However, the approximately close biomass production appears by the model is an expectable observation under the conditions of simulation. Besides the uncertainty caused by additional model parameters used in the biocrust model, there also be an added uncertainty caused by the model structure and simplification assumptions is not considered in this study.

The increasing biomass within the biocrust stems from a reaction-diffusion interaction. The model predicts the biomass concentration gradient as well as the soil water content as a result of suction gradient is established by biomass water uptake as the cell began to growth. This water gradient forces the water to move towards domain's centre from the boundaries. This process continuous as long as the domain's soil has enough water storage. Biomass concentration is increasing near the biocrust interface and becomes reduced in the biocrust interior. However its distribution is homogeneous in the domain x and y directions surface. The profile of viable cell fraction is reversed in which cell numbers is in higher numbers of the biocrust surface and are decrease effectively near the domain boundaries. Viable cell numbers and biocrust thickness reduction were dissimilar measures of light efficacy. The general behaviour predicted by the model is in good qualitative agreement with an experimental observations. As the initial seeding was considered similar in domain, therefore, biomass distribution would be homogeneous in all of the surface.

Fig. 2 The solution of biomass partial differential equation with two Dirichlet and two Neumann boundaries (and initial condition of 0.03). Results of the temporal and spatial biomass concentration distribution profile in the x and y surface. (a) Temporal distribution, and (b) spatial distribution with surface plot.

The modelling of soil water is more complex as it is not only dependent to up taken by microorganisms, but also dependent to forces acting on the soil water mass balance such as soil porespaces, evapotranspiration, and input-output water flow to soil. If we assume no input-output water flow, and water is free for movement in the soil, the model prediction reveals of a logarithmic water reduction curve during cultivation period. The decreasing speed in this case is most inversely corresponds to cell growth rate. Figure 3 reveals time profile of soil water in the domain during cultivation period in the horizontal surface.

٢٢٥ Fig. 3 The solution of diffusion partial differential equation of soil water content with two Dirichlet and two Neumann boundaries (and initial condition of 0.55). Results of the soil water content 
[revised manuscript text omitted]

Shackelford, C. D. and Daniel, D. E.: Diffusion in saturated soil. I: Background, J. Geotech. Eng., 117(3), 467–484, 1991.

Systems, T.: Systems and control group identification of algae growth kinetics, Wageningen. [online] Available from:

http://edepot.wur.nl/9247, 2009.

Vasilyeva, N. A., Ingtem, J. G. and Silaev, D. A.: Nonlinear dynamical model of microorganism growth in soil, Comput. Math. Model., 27(2), 172–180, doi:10.1007/s10598-016-9312-7, 2016.

Vidriales-Escobar, G., Rentería-Tamayo, R., Alatriste-Mondragón, F. and González-Ortega, O.: Mathematical modeling of a composting process in a small-scale tubular bioreactor, Chem. Eng. Res. Des., 120, 360–371, doi:10.1016/j.cherd.2017.02.006, 2017.

٣٥٥

Table 1

| BSC | Biofilm |
|---|---|
| Has higher species diversity | Has single or few species diversity |
| The associated communities are bacteria, fungi, lichen and mosses | The associated communities are similar microorganisms e.g. bacteria |
| In addition to cyanobacteria, other microorganisms such as fungi participate in EPS production. | Cyanobacteria are main EPS producers |
| Coral and fine particles are main components | Only biofilm components are constituent species |
| They are more porous | They are less porous |
| *Microcoleus* lack UV protecting pigments, then stratifies in the sub-surface, while some other such as *Nostoc* having UV screening pigments stratify at the soil surface. | Relatively homogeneous |
| The substratum cannot be defined | The substratum is fixed flat plate |
| There is no detachments, and the decayed cell remain in the soil. | The cell detached from the biofilm surface. |

Table 2

| Assumed condition | Real condition |
| --- | --- |
| Two-dimensional (2D) - BSC structure parallel to soil surface | Three-dimensional (3D) biocrust structure |
| Homogenous biofilm structure, (uniformly thick planar aggregate) | Heterogeneous biocrust structure (physical, chemical, biological and geometrical) |
| Biocrust is solid matrix and rigid structure | Biocrust is multiphase system |
| Pure species biocrust | Diverse multispecies biocrust |
| Fixed biocrust boundary conditions | Moving biocrust boundary conditions |
| Biocrust is flat | Biocrust is fractal shape |
| Biofilm thickness is constant. | Biocrust volume and thickness is variable |
| Diffusion rate of substrate is constant | Diffusion rate of substrate is variable |
| Temperature is constant and in optimum conditions | There is temperature fluctuation in the biocrust |
| All variables over areas parallel to the substratum are constant. | Variables are not uniform over surface area parallel to substratum |
| Light kinetics describe by DiToro et al kinetics | Kinetic of light variation is complex inside of the biocrust |
| Substrate concentration and environmental parameters are not limiting except light illumination and water | Many substrate and environmental parameters are limiting and complex inside of the biocrust |
| Light enters the biocrust domain perpendicular to surface | Light angle and intensity change with time and locations |
| The domain boundaries are no diffusible and fix | The biocrust boundaries are not fix and growing fractal. |

Table 3

| Name | Symbol | Unit | Value | Ref. |
|------|--------|------|-------|------|
| Biomass distribution coefficient | $D_B$ | $m^2/d$ | 6.25E-4 | a |
| Maximum specific growth rate | $\mu_{max}$ | $d^{-1}$ | 0.4 | b |
| Water distribution coefficient | $D_w$ | $m^2/d$ | 6.25E-2 | a |
| Effective water diffusion coefficient | $D_w^*$ | $m^2/d$ | | d |
| Retardation factor | $R_d$ | - | | d |
| Optimal light illumination coefficient | $I_s$ | $\mu$mol photone/$m^2\cdot d$ | 0.007 | a |
| Michelis – Menton coefficient | $K_w$ | W/W | 0.01 | b |
| Soil porosity $(V_V/V_T)^*$ | n | % | 0.3-0.7 | c |
| Soil water content (mass percent = $m_{biomass}/m_{soil \times 100 \ (W/W)}$) | V | W/W | 0.2 | This study |
| Time | t | d | 60 | This study |

a, (DiToro et al., 1971); b, (Systems, 2009); c,(Schiavone, 2016); d, (Shackelford and Daniel, 1991).

$V_V$ is the volume of void-space (such as fluids) and $V_T$ is the total or bulk volume of material, including the solid and void components.

Fig. 1

[Figure]

**a**

[Figure]

Fig. 3

**a**

[Figure]

[Figure]